# A Multifield Study on Dimethyl Acetylenedicarboxylate: A Reagent Able to Build a New Cycle on Diaminoimidazoles

**DOI:** 10.3390/molecules27103326

**Published:** 2022-05-22

**Authors:** Dmitrii Yu. Vandyshev, Oleg N. Burov, Anton V. Lisovin, Daria A. Mangusheva, Mikhail A. Potapov, Tatiana N. Ilyinova, Khidmet S. Shikhaliev, Athina Geronikaki, Domenico Spinelli

**Affiliations:** 1Department of Organic Chemistry, Chemical Faculty, Voronezh State University, Universitetskaya Sq. 1, 394018 Voronezh, Russia; mangusheva.dasha@yandex.ru (D.A.M.); amidines@mail.ru (M.A.P.); shikh1961@yandex.ru (K.S.S.); 2Department of Chemistry, Southern Federal University, Bolshaya Sadovaya Str. 105/42, 344090 Rostov-on-Don, Russia; bboleg@gmail.com (O.N.B.); tonnie.fox@gmail.com (A.V.L.); 3Department of Clinical Laboratory Diagnostics, Voronezh Medical State University, Studencheskaya Street 12, 394036 Voronezh, Russia; tatiana_ilinova@mail.ru; 4School of Pharmacy, Aristotle University of Thessaloniki, 54124 Thessaloniki, Greece; 5Independent Researcher, Via Marconi 32, 40122 Bologna, Italy

**Keywords:** dimethyl acetylenedicarboxylate, imidazo[1,5-*b*]pyridazines, DFT calculations

## Abstract

A new effective method for the synthesis of imidazo[1,5-*b*]pyridazines derivatives (yields = 68–89%) by the interaction of 1,2-diamino-4-phenylimidazole with DMAD, in methanol and in the presence of a catalytic amount of acetic acid, is proposed. The course of reaction has been examined by classical organic methods, HPLC-MS analysis, and quantum-chemical calculations.

## 1. Introduction

Several natural biologically active systems (from chlorophyll to hemoglobin) and many natural and synthetic drugs (from penicyllin to sulfadiazine) contain in their structure heterocyclic systems (mono- or poly-cyclic) holding different hetero-atoms (from nitrogen to oxygen or sulphur).

This is likely one of the reasons that so many chemists study the syntheses, the reactivities, andthe properties of several heterocyclic compounds.

Thus, in the past fewdecades we have studied rearrangements [1,2,3,4], heterocyclicring-openings [5,6,7,8,9], and the different behaviors [10,11,12] of many heterocyclic compounds, also gaininguseful information on their pharmacological activities (cardiovascular [13,14,15], antitumor [16,17,18], and anti-MDR1 [19,20] effectshave been evaluated).

We have focusedour attention onboth mono- and poly-cyclic systems. Now, we are turning our interest to the study of a reaction able to "build" a new ring on an already existing one.

Among the reagents employed in this field, special attention has been devoted to the use of dimethyl acetylenedicarboxylate (**1**, DMAD). It represents a versatile synthetic reagent useful for the discovery of new methods in combinatorial chemistry and multicomponent reactions, able to produceheterocyclic compounds. Being an electron-deficient acetylene derivative, it can act as a Michael acceptor or participate in cycloaddition reactions (Diels-Alder, 1,3-dipolar and [2 + 2] cycloaddition) depending on the chosen reagents/conditions. The ability of DMAD to form zwitterions in multicomponent processes is quite important. The interactions of DMAD with amino azoles, leading to the formation of different heterocyclic systems from structurally similar initial bi-nucleophilic substrates, depend on experimental conditions and on reagents used. In fact, the observed different results can depend onthe used solvent, onthe catalyst, and onthe nature of the substituents present in the used substrates.

For example, it was found that *N*,*N*- and *N*,*S*-dinucleophiles, such as amino- and mercaptotriazoles **2a** and **2b**, respectively [21,22], by interaction with DMAD in dichloromethane, led to the formation of 4,7-dihydro-s-triazolo[1,5-a]pyrimidinones **4a**, [1,2,4]triazolo[5,1-b][1,3]thiazinones **4b** and dihydro[1,2,4]triazolo[4,3-*a*]pyrimidinones **4c** with high antibacterial activity [23,24] (Figure 1). In a first step, an addition ofDMAD at the multiple bond by the amino or mercapto group of the starting triazole occurs, then an intramolecular cyclization on the NH fragment due to the “far” ester group happens. A similar process, involving *N*-substituted aminotriazoles **6**, [25] aminoimidazoles **7** and **10**, [26,27] aminopyrazoles **3**, [28] 2-amino-3-carboethoxyindoles **12b** [29], and indoline-2-thiones **12a** [30] leads to the formation of [1,2,4]triazolo[4,3-*a*]pyrimidinones **8**, imidazo[1,2-*a*]pyrimidinones **9** and **11**, pyrazolo[1,5-*a*]pyrimidinones **5**, pyrimido[3,2-*a*]indolones **14**, and thiopyrano[2,3-*b*]indolones **13**.

With some dinucleophiles, the intermediates initially formed by the additionon the multiple bonds of DMAD can undergo cyclization at the nearest ester group. Such reaction has been described for 2-hydrazinobenzimidazoles **15** [31] and 3,4-dihydropyrimidine-2-thiones **16a**–**c**, [32,33,34] with the formation of dihydrobenzo[4,5]imidazo[2,1-c][1,2,4]triazines **17** and dihydrothiazolo[3,2-*a*]pyrimidines **18**, respectively (Figure 2).

Ueda et al. [35] found that when DMAD was reacted with 7-alkyl-8-amino-1,3-dimethylxanthines amines **19** in methanol, 5-alkyl-1,3-dimethyl-9-methoxycarbonyl-1,2,3,4-tetrahydro-5*N*,7*N*-pyrimido[1,2-*e*]purine-2,4,7-triones **20** and 7-alkyl-1,3-dimethyl-1,2,3,6-tetrahydro-8-(1,2,4,5,6,7-hexamethoxycarbonyl-3-azatricyclo[2.2.1.0^2,6^]heptyl)purine-2,6-diones **21** were formed (Figure 3). However, the reaction, performedin dimethylformamide, led to the formation of the mixture of diazocino- and diazepinopurines **22**–**24**.

It has also been observed that 1,3-substituted 5-aminopyrazole-4-carbonitriles **25** [36] (in the presence of potassium carbonate, in DMSO) and *N*,*N*-diisopropylamidinylpyrazolimines **26** [37] (in the presence of catalytic amounts of zinc chloride in acetonitrile) interact with DMAD (in case **26** DEtAD **1*** has been used), inducingan Aza-Diels–Alder reaction, leading to the formation of the relevant 4-aminopyrazolo[3,4-*b*]pyridine-5,6-dicarboxylates **27** and 1*H*-pyrazolo[3,4-*b*]pyridine-4,5-dicarboxylates **28** (Figure 4), respectively.

It was discovered that when1-substituted 5-aminopyrazoles **30** (prolonged boiling in acetic acid), [38] 1-methyl-2-aminoindoles **29** (prolonged boiling in methanol), [29] 3-aminocoumarin **31**, 3-aminochromone **32** (in methanol and diphenyl ether) [39], and aminouracil **33** (methanol at room temperature) [40,41] were used, DMAD acts as a Michael acceptor, promoting the formation of the corresponding heterocyclic systems (Figure 5).

Reactions with polynucleophiles are quite complex. In fact, Miyamoto [42] observed that the course of the reaction of DMAD **1** with 2-amino-4-R-1-arylideneaminoimidazoles **39** in benzene depends on the nature of the present substituents and the temperature of the reaction. For example, interaction with 2-amino-1- benzylidenamino-4-methylimidazole **39a** in boiling benzene led to the formation of dimethyl-2-amino-1-benzylideneamino-1*H*-pyrrole-3,4-dicarboxylate **40a** and methyl 1-benzylidenamino-4-methyl-5-oxo-1*H*-imidazole[1,2-*a*]pyrimidine-7-carboxylate **41**. Vice versa, in the instance of 2-amino-4-aryl-1-arylideneaminoimidazoles **39b**, the previously described pyrroles **40b** in a mixture with benzonitriles **42** were isolated after boiling. Moreover, performing the reaction at room temperature the formation of dimethyl-1- (2-amino-1-benzylideneamino-4-aryl-1*N*-imidazol-5-yl)fumarates **43** was observed (Figure 6).

Looking at the complex aforementioned results, we have focusedour attention onthe study of the reaction of DMAD with non-symmetric polynucleophiles deriving from 1,2-diaminoimidazoles: the influence of reaction conditions on the regioselectivity will be examined using classical organic synthesis and quantum chemical calculations.

## 2. Results and Discussion

Considering the fact that the observed course of reaction of DMAD **1** with some nucleophiles depends on the nature of the solvent, first we have investigated and selected the optimal reaction conditions, examining the possible interactions of **1** with 1,2-diamino-4-phenylimidazole **44a**. Based on the literature data and considering the solubility of the starting diamine, we have chosen thepossible solvents benzene, dioxane, chloroform, methylene chloride, methanol, and ethanol. The composition of the reaction mixture was assessed using the absolute normalization method based on HPLC-MS analysis. The signals were interpreted on the basis of preliminary calculated weights, in the form of molecular ions with [M+H]^+^ of all the possible initial, intermediate, and expected final compounds. Samples were treated for 0, 60, and 120 min. It should also be noted that under the conditions of our HPLC-MS experiment, DMAD was not protonated or detected at a given wavelength. The results obtained are portrayedin Table 1.

As can be observed, when solvents such as benzene, dioxane, chloroform, or methylene chloride were used, the highest conversion of the starting diamine after 2 h of boiling is about 37–65%. In this case, the mass spectra of the reactionsindicatesignals of the formed intermediates and products of their intramolecular cyclization.

A different behavior was observed when methanol or ethanol were used as solvents: complete conversion of 1,2-diamino-4-phenylimidazole occurred in 2 h. However, using ethanol, the formation of the target reaction product was significantly low, in line with an ongoing process of transesterification of the ester group. Moreover, the formation of a mixture of ethyl and methyl esters makes more complex the identification/analysis of the products of reaction.

We have previously observed that heterocyclization reactions involving diaminoimidazole are accelerated by acid catalysis (for example, by acetic acid) [43,44,45]. In this case, we have observedthat the addition of acetic acid, using methanol as a solvent significantly affected the reactivity. In fact, as presented in Table 1, the addition of catalytic amounts of acetic acid allows a complete conversion of the reactant in 60 min. It should be noted that, on the contrary, conducting a reaction in a mixture of acetic acid and methanol (1:1) reduces the reactivity of the reagents and leads to the formation of several by-products.

Thus, it was discoveredthat the reaction between dimethylacetylenedicarboxylate **1** and 1,2-diamino-4-phenylimidazoles **44a**–**e** proceeds most smoothly when methanol with the addition of a catalytic amount of acetic acid was used as solvent. In this case, the complete conversion of the starting diamine and the maximum yield of the target product were achieved after an hour of boiling the reagents.

Considering the polynucleophilic nature of the starting 1,2-diaminoimidazoles **44a**–**e**, Figure 7 reports the structures of all possible intermediates of their interaction with DMAD **1**. Thus, during the first step, the nucleophilic addition of **44a**–**e** on the triple bond of **1** due to the nucleophilic CH, NH, and NH_2_ groups, in addition to the nitrogen atom of the cycle, can lead to the formation of the intermediates **A**–**D**. The resulting Michael adducts can undergo intramolecular cyclization, due to the “near” or “far” carboxyl group, with the formation of the corresponding products **45**–**50**.

To gain information on the real course of the reaction, we have examined the ^1^H-NMR spectra of the products of reaction. We have observed that, in them, the characteristic CH signals of the imidazole ring fragment and of the amino group of the hydrazine fragment were absent; this observation allowed us to exclude the possibility that compounds **49** and **50** were formed. A further confirmation of this hypothesis was derived from the presence of the amino group signals at C-7 (6). Moreover, the spectra of the isolated compounds contained signals of the protons of the pyridazine ring at δ = 5.9–6.3 ppm, while the spectra of compound **45a**, in which R was a hydrogen atom, contained the characteristic signal of the amide proton at δ = 11.75 ppm. Thus, it was possible to also exclude the structures **46** and **48**. At this point, we can confirmthat only the structures **45** and **47** are possible for the obtained products of reaction.

The key criterion for choosing between these two structures (**45** and **47**) was the presence of cross peaks in the NOESY spectrum between the protons of the methyl group of the ester fragment and the *ortho*-protons of the aromatic ring present on C-5 of the imidazole ring, as presentedin Figure 1 for compound **45b**. Thus, we can conclude that the reaction of1,2-diaminoimidazoles **44a**–**e** with DMAD produces the methyl esters of 7-amino-1,2-dihydro-1-R-2-oxo-5-phenylimidazo[1,5-*b*]pyridazine-4-carboxylic acids **45a**–**e**.

Of course, the structures of all the obtained compounds (**45a**–**e**) have been confirmed by the collected spectroscopic data (^1^H- and ^13^C-NMR, in addition to HRMS data). All of the examined reactions occurred with good/excellent yields (68–89%).

To obtainan additional confirmation of the structure of the obtained imidazopyridazines **45a**–**e** and as a further step of our research, we have carried out the transformation of methyl 7-amino-1,2-dihydro-2-oxo-5-phenylimidazo[1,5-*b*]pyridazine-4-carboxylate **45a** into 7-amino-5-phenylimidazo[1,5-*b*]pyridazin-2-one **52** (Figure 8). Thus, by alkaline hydrolysis **45a** producedthe relevant carboxylic acid **51**, which, in turn, by refluxing in diphenyl ether in the presence of copper acetate, furnished by decarboxylation (pathway A) the 7-amino-5-phenylimidazo[1,5-*b*]pyridazin-2-one **52**. The structure of isolated compounds **51** and **52** was proven by ^1^H- and ^13^C-NMR spectroscopy and HRMS.

The structure of imidazopyridazine **52** was also confirmed by the NOESY spectrum, in which cross-interaction between the CH proton of the pyridazine ring and *ortho*-protons of the aromatic ring was observed (Figure 2). This kind of interaction is impossible for this couple of protons in all alternative structures **46**–**48**. Therefore, the proven structure of imidazopyridazine **52** has further confirmed that in the course of the interaction between 1,2-diaminoimidazoles **44a**–**e** and DMAD, methyl esters of 7-amino-1,2-dihydro-1-R-2-oxo-5-phenylimidazo[1,5-*b*]pyridazine-4-carboxylic acids **45a**–**e** have been obtained.

The structure of compound **52** was further confirmed by preparing it with an alternative synthesis, id est by treating the 1,2-diamino-4-phenylimidazole **44a** with ethyl propargylate **53** (Figure 8, pathway B). As in the instance of the reaction with DMAD, the optimal condition was boiling for 60 min the mixture of reagents in methanol with the addition of a couple of drops of acetic acid under constant stirring.

To obtainfurther information on the course of the reaction between 1,2-diaminoimidazoles **44** and DMAD **1**, we have calculated the minimum energy paths (MEP) for the reaction. Quantum-chemical DFT calculations were performed using B3LYP/6-311++G(d, p) basis and considering the solvation effects by the polarizable continuum model (PCM) in methanol.

Firstly, we examined the interaction of the simplest 4-phenyl-1*H*-imidazole-1,2-diamine **44a** and DMAD **1**, leading to the formation of **45a**.

In the first step of this process, anucleophilic attack of carbon atom C-5 of imidazole on a carbon atom of the triple bond of DMAD **1** occurred with the formation of the bipolar σ-complex **54**, which was 23.3 kcal/mol higher in energy comparedto the starting reagents (Figure 3).

Now we will discuss the change in the relative free energy ΔG. During the second step of the process, a 1,3-hydrogen shift occurred and the covalent σ-complex **55** was formed: although the formation of this structure required an activation energy of 21.0 kcal/mol, **55** is produced, which is much more stable. In the final step, an intramolecular nucleophilic attack by the nitrogen atom of the amino group on the carbonyl carbon atom occurred, followed by the elimination of the methanol molecule and the formation of the final product **45a**. The formation of **45a** was highly exothermic (a gain of 30.6 kcal/mol occurred), and it required the overcoming of a very large energy barrier (45.5 kcal/mol) in the last step (**TS3**) (Figure 9).

Then we considered the reaction proceeded via the intermediate D, theoretically leading to the formation of product **50-2** (Figure 7 and Figure 10).

During the first step of the studied process, a nucleophilic attack by N-3 of imidazole on a carbon atom of the triple bond of DMAD **1** occurred with the formation of the bipolar σ-complex **56**. As in the previous instance, the formation of a non-classical structure is energetically unfavorable; in fact, structure **56** was 20.2 kcal/mol higher in energy than the initial reagents (Figure 4). During the second step, a 1,5-hydrogen shift occurred and the compound **57** was formed. It should be noted that **57** appears to be the final product; in fact, further cyclization into product **50-2** was not observed. In general, this path was weakly endothermic: **57** was energetically less favorable than the initial reagents, then the formation of complex **D** was unlikely.

Now the formation of a complex of type **C** (leading to products **49-2** or **50-1**) will be examined (Figure 7 and Figure 11).

Now, during the first step of the reaction, anucleophilic attack of the nitrogen atom of the amino group linked at the C-2 atom of imidazole occurred on a carbon atom of the triple bond of DMAD **1** with the formation of the bipolar σ-complex **58**: it was 33.4 kcal higher in energy than the starting reagents (Figure 5). The following 1,3-hydrogen shift led to the formation of the covalent adduct **59**. However, as in the previously examined process (Figure 10), the following cyclization into products **49-2** or **50-1** did not exist on the potential energy surface (PES).

The processes involving the formation of type **B** complexes proceeded through the formation of a bipolar σ-complex. Zwitterionic adduct **60** was formed by the nucleophilic attack of the *N*-amino group of imidazole **44a** on a carbon atom of the triple bond of DMAD **1** (Figure 12), then giving the covalent adduct **61** as a result of the 1,3-hydrogen shift. The formation of adduct **61** from DMDA **1** and **44** was energetically favorable and was accompanied by an energy gain of 17.3 kcal/mol (Figure 6). The maximum barrier of the process **1** + **44**→**61** concerns the first stage (**TS8**) and required an activation energy of 38.2 kcal/mol. As in the two previous cases, the formation of a cyclic structure from adduct **61** was not suggested by PES.

In conclusion, calculations for the bimolecular pathways of interaction of 4-phenyl-1*H*-imidazole-1,2-diamine **44a** and DMAD **1** indicatedthat between the hypothetical cyclic products **45**–**50**, only the formation of a cyclic system **45a** appeared as a possible path of reaction. In addition, data of quantum-chemical calculations on the course of the bimolecular processes indicatedthat the open-chain covalent adducts **59** and **61** could be formed, but their formation occurred with an energy barrier in the first stage higher than in the instance of the formation of adduct **54**.

Since, according to the results of the experiments carried out using an alcohol as solvent of reaction, re-esterification products were recorded, we concludedthat the interaction of 4-phenyl-1*H*-imidazole-1,2-diamine **44a** and DMAD **1** can develop not only via a bimolecular process, but also as a trimolecular process with the involvement of an alcohol molecule. For this, we performed the quantum-chemical calculation of paths involving methanol.

Firstly, we have considered the trimolecular process concerning the formation of the target product **45a** (Figure 13, Figure 7). At the molecular level, the methanol molecule affects all steps of the process. Thus, in the first step, during the formation of the bipolar σ-complex **62**, the proton of the hydroxyl of methanol was coordinated to the carbon atom charged negatively inthe multiple bond. However, no stabilization of the zwitterionic structure (compared with the process on Figure 9) occurred; in fact, the adduct **62** was less energy-efficient by 27.1 kcal/mol compared to the starting reagents. Furthermore, the lowbarrier of the 1,3-hydrogen shift which led to **63** was performed through a five-center transition state (**TS11**), in which the molecule of methanol acted as a bifunctional catalyst.

Finally, the covalent **63** was transformed into product **45a** through the transition state **TS12**, in which the molecule of methanol again performed the function of a bifunctional catalyst and participated in the formal transfer of a proton from the *N*-amino group to the ester oxygen atom. Despite the participation of the methanol in the proton transfer, the formation of product **45a** from **63** proceeded with a barrier of 48.2 kcal/mol, which was higher than in the process portrayedin Figure 9.

The addition of a methanol molecule in the process presented in Figure 10 causes an effect at the molecular level (Figure 14). Thus, when the bipolar intermediate **64** was formed from the initial reagents, the methanol coordinated the proton of the hydroxy group to the carbon atom of the double bond.

Despite the coordination of the zwitterionic **56** with a molecule of methanol in complex **64**, this was 28.1 kcal/mol higher in energy than the initial reagents, which in turn means 7.9 kcal/mol higher in energy than for **56** in Figure 10 (compare Figure 4 and Figure 8).

The methanol molecule facilitated the 1,5-proton transfer, being included in the eighth-center transition state (**TS14**), performing the function of a bifunctional catalyst. As before, the process of the formation of structure **57** was endothermic.

When the methanol molecule was involved in the processes reported in Figure 11 and Figure 12, we observed a similar pattern (see Appendix A). It was coordinated with the zwitterion, destabilizing it. However, during the following step of the process, the methanol acted as a bifunctional catalyst facilitating the transfer of the hydrogen atom. In this case, the reactions, as in the case of the bimolecular pathway, stop at the stage of the formation of the covalent adducts.

According to the calculation data, a pattern analogous to the trimolecular processes with the involvement of methanol was observed in the case of the participation of acetic acid in the trimolecular process at the molecular level. Thus, for the process of formation of the compound **45a** from the initial compounds, the barrier of the first stage of the formation of the zwitterionic associate was 40.4 kcal/mol, and the zwitterionic associate itself was 26.3 kcal/mol higher in energy than the initial reagents. The covalent adduct into which it rearranged, as a result of electron transfer, was 9.0 kcal/mol higher in energy comparedto the starting reagents. The trimolecular process with the involvement of acetic acid, as in the case of the participation of methanol, TS was higher in energy than in the corresponding bimolecular processes. Obviously, in all the cases considered, the involvement of a polar protic solvent at the molecular level can make worse the course of reactions. However, the covalent and zwitterionic intermediates in trimolecular processes were higher in energy than the intermediates infinitely separated from the solvent molecules, which means that the reaction can proceed by bimolecular pathways without switching to the trimolecular ones. Then, the experimentally observed ease of processes in polar solvents is explained exclusively by the dielectric constant (polarity of the medium). In fact, the higher the polarity of the medium, the higher the stabilizationenergy of zwitterionic or ionic processes [46].

When switching from the gas-phase calculations to the PCM medium accounting for the process in Figure 9, we observed a reduction in the barrier values. Thus, the barrier of the first stage (TS1) is only 27 kcal/mol (see Appendix A) and the thermodynamic gain of the whole process increased to 39.3 kcal/mol.

Thus, based on quantum chemical calculations, the following conclusions can be drawn:-the predominant cyclic product of the interaction of diaminoimidazoles **44** with DMAD **1** is imidazopyridazines **45**;-open-chain nucleophilic addition products of diaminoimidazoles **44** to DMAD **1** at positions other than the C-2 carbon atom, which are less preferable kinetically, can participate in a further cascade of processes leading to the formation of difficultly separable mixtures not involved in further intramolecular cyclization;-interaction of diaminoimidazoles **44** with DMAD **1** is complicated in the instance that methanol or acetic acid participate to the reaction. In fact, the participation of the molecules of the third component in the formation of the primary bipolar σ-complexes can destabilize them. Conversely, during the following steps of the process, they can facilitate the transfer of protons because of the formation of classical covalent adducts;-the fact that all stationary points in the instance of trimolecular processes are higher in energy than the sum energies of the corresponding points and solvent molecules in case of bimolecular processes makes trimolecular processes alternative, but not real, processes.-polar solvents can facilitate the formation processes of type **45** systems due to their polarity (ε value), stabilizing polar intermediates, and not due to their direct involvement in the processes.

## 3. Materials and Methods

### 3.1. General

^1^H NMR, ^13^C NMR, NOESY spectra were recorded on BrukerDRX-500 devices (500.13 and 125.75 MHz, respectively) in DMSO-d_6_ and TFA-d with internal TMS standard. Melting points were taken on a Stuart SMP30 device (Cole-Parmer Ltd., St. Neots, UK). HPLC/MS spectra were recorded on an Agilent Infinity 1260 chromatograph (Agilent Technologies, Palo Alto, CA, USA) with MS interface Agilent 6230 TOFLC/MS. Conditions for the separation: mobile phase MeCN/H_2_O + 0.1% FA (formic acid), gradient elution (first CH_3_CN:H_2_O (60:40), then for 5 min to 90% CH_3_CN), column—Poroshell 120 EC-C18 (4.6 × 50 mm, 2.7 μm), thermostat 23–28 °C, flow rate of 0.3–0.4 mL/min electrospray ionization (capillary—3.5 kV; fragmentor +191 V; OctRF +66 V—positive polarity). The course of the reaction and the purity of the obtained compounds were controlled by TLC on Merck TLC Silica gel 60 F254 plates in a 20:1 CHCl_3_–MeOH system (visualization under UV light). The commercially available reagents were purchased from Acros Organics (Geel, Belgium).

Quantum chemical DFT calculations were carried out in the 6–311++G(d,p) triple-zeta basis using a B3LYP functional [47]. This basis has provideda good account of itself in reproduction of vibrational frequencies, geometry, and minimum energy reaction paths [48,49,50]. Full geometry optimization of molecular structures corresponding to stationary points of the potential energy surface was carried out as high as a gradient value of 10^−7^ hartree/bohr according to the Gaussian 09 software package (Wallingford, CT, USA) [51] using the Silver cluster of the Research Institute of Physical and Organic Chemistry at SFedU. The nature of stationary points was determined relying on the calculation of normal vibration frequencies (the Hessian matrix). MEPs were plotted using gradient descent from transition states in forward and reverse directions of transition vectors. In order to search for transition states, linear and quadratic synchronous transit methods were used [52,53].

### 3.2. General Procedure for the Synthesis of Methyl 7-Amino-1,2-dihydro-1-R-2-oxo-5-phenylimidazo[1,5-b]pyridazine-4-carboxylates (***45a***–***e***)

Acetylene dicarboxylic acid dimethyl ester **1** (5 mmol) was added dropwise to 5 mmol of diaminoimidazole **44a**–**e** dissolved in 5 mLof methanol containing catalytic amounts of acetic acid (2–3 drops). After adding **1**, the reaction mixture was refluxed for 60 min. The precipitated crystals were filtered off and recrystallized from MeOH.

#### 3.2.1. Methyl 7-Amino-1,2-dihydro-2-oxo-5-phenylimidazo[1,5-*b*]pyridazine-4-carboxylate (**45a**)

Yield (124.4 mg, 89%), orange. Mp:239–241 °C.^1^H NMR (500 MHz, DMSO-*d*_6_, δ ppm): 11.75 (br.s, 1H), 7.35–7.39 (m, 2H), 7.26–7.32 (m, 3H), 6.23 (s, 1H), 5.91 (s, 2H), 3.41 (s, 3H). ^13^C{^1^H} NMR (125.8 MHz, DMSO-*d*_6_, δ ppm): 164.9, 158.4, 143.7, 135.4, 134.1, 129.2, 127.8, 127.6, 126.7, 111.0, 104.5, 52.2. HRMS (ESI) *m*/*z*: [M+H^+^] calc. for C_14_H_12_N_4_O_3_, 285.0982, found 285.0980.

#### 3.2.2. Methyl 7-Amino-1-benzyl-1,2-dihydro-2-oxo-5-phenylimidazo[1,5-*b*]pyridazine-4-carboxylate (**45b**)

Yield (127.1 mg, 68%), orange. Mp: 153–155 °C. ^1^H NMR 500 MHz, DMSO-*d*_6_, δ ppm): 7.30–7.38 (m, 3H), 7.23–7.29 (m, 3H), 7.20 (d, *J* = 7.2, 2H), 7.09 (d, *J* = 7.2, 2H), 6.71 (s, 2H), 5.97 (s, 1H), 5.42 (s, 2H), 3.27 (s, 3H). ^13^C{^1^H} NMR (125.8 MHz, DMSO-*d*_6_, δ ppm): 164.7, 164.5, 146.8, 137.4, 135.5, 135.4, 133.9, 128.6, 128.1, 128.0, 127.9, 127.7, 112.5, 108.9, 52.3, 47.8. HRMS (ESI), *m*/*z*: [M+H^+^] calc. for C_21_H_18_N_4_O_3_, 375.1453, found 375.1452.

#### 3.2.3. Methyl 7-Amino-1-(3-chlorobenzyl)-1,2-dihydro-2-oxo-5-phenylimidazo[1,5-*b*]pyridazine-4-carboxylate (**45c**)

Yield (145 mg, 71%), yellow. Mp: 237–239 °C. ^1^H NMR (500 MHz, DMSO-*d*_6_, δ ppm): 7.30–7.40 (m, 5H), 7.21–7.24 (m, 2H), 7.12 (s, 1H), 7.01–7.04 (m, 1H), 6.73 (s, 2H), 5.99 (s, 1H), 5.42 (s, 2H), 3.28 (s, 3H). ^13^C{^1^H} NMR (125.8 MHz, DMSO-*d*_6_, δ ppm): 164.5, 164.3, 146.7, 137.7, 137.5, 135.5, 133.8, 133.0, 130.5, 128.0, 128.0, 129.9, 127.5, 126.1, 112.4, 108.8, 52.2, 47.2. HRMS (ESI), *m*/*z*: [M+H^+^] calc. forC_21_H_17_ClN_4_O_3_409.1061, found 409.1060.

#### 3.2.4. Methyl 7-Amino-1-(2-methoxybenzyl)-1,2-dihydro-2-oxo-5-phenylimidazo[1,5-*b*]pyridazine-4-carboxylate (**45d**)

Yield (175.7 mg, 87%), red. Mp: 185–187 °C.^1^H NMR (500 MHz, DMSO-*d*_6_, δ ppm): 7.29–7.38 (m, 3H), 7.17–7.21 (m, 3H), 7.13 (dd, *J* = 1.65, *J* = 8.5, 1H), 6.87 (d, *J* = 7.9, 1H), 6.79 (t, *J* = 7.5, 1H), 6.64 (s, 2H), 5.89 (s, 1H), 5.30 (s, 2H), 3.64 (s, 3H), 3.29 (s, 3H). ^13^C{^1^H} NMR (125.8 MHz, DMSO-*d*_6_, δ ppm): 165.2, 164.7, 157.6, 147.3, 137.3, 135.1, 134.1, 130.7, 129.5, 127.9, 127.8, 127.8, 122.3, 119.7, 112.3, 110.2, 108.9, 54.9, 52.2, 46.8. HRMS (ESI), *m*/*z*: [M+H^+^] calc. for C_22_H_20_N_4_O_4_ 405.1558, found 405.1557.

#### 3.2.5. Methyl 7-Amino-1-(4-methylbenzyl)-1,2-dihydro-2-oxo-5-phenylimidazo[1,5-*b*]pyridazine-4-carboxylate (**45e**)

Yield (149.4 mg, 77%), orange. Mp: 170–172 °C. ^1^H NMR (500 MHz, DMSO-*d*_6_, δ ppm): 7.30–7.38 (m, 4H), 7.20 (d, *J* = 7.2, 1H), 7.07 (d, *J* = 7.8, 2H), 6.98 (d, *J* = 7.7, 2H), 6.68 (s, 2H), 5.96 (s, 1H), 5.37 (s, 2H), 3.27 (s, 3H), 2.21 (s, 3H). ^13^C{^1^H} NMR (125.8 MHz, DMSO-*d*_6_, δ ppm): 164.5, 164.4, 146.7, 137.2, 137.2, 135.3, 133.9, 132.3, 129.1, 128.0, 127.9, 127.9, 127.6, 112.4, 108.9, 52.2, 47.3, 20.5. HRMS (ESI), *m*/*z*: [M+H^+^] calc. for C_22_H_20_N_4_O_3_ 389.1609, found 389.1608.

### 3.3. Synthesis of 7-Amino-1,2-dihydro-2-oxo-5-phenylimidazo[1,5-b]pyridazine-4-carboxylic Acid (***51***)

Potassium hydroxide (5.5 mmol),dissolved in 5 mLof water, was added to 5 mmol of **45a**. The mixture was boiled for 2 h, until **45a** completely disappeared (control by TLC). After cooling, the mixture was acidified with hydrochloric acid. The formed precipitate was filtered off.

#### 7-Amino-1,2-dihydro-2-oxo-5-phenylimidazo[1,5-*b*]pyridazine-4-carboxylicAcid (**51**)

Yield (124.2 mg, 92%),lightyellow. Mp: >300 °C. ^1^HNMR (500 MHz, DMSO-*d*_6_,δppm): 11.72 (br.s, 1H), 7.40–7.43 (m, 2H), 7.29–7.34 (m, 3H), 7.22 (t, *J* = 7.3, 1H), 6.17 (s, 1H), 5.94 (s, 2H). ^13^C{^1^H} NMR (125.8 MHz, DMSO-*d*_6_, δ ppm): 166.3, 158.8, 143.3, 137.1, 134.6, 127.8, 127.5, 126.5, 111.3, 103.9. HRMS (ESI), *m*/*z*: [M+H^+^] calc. for C_13_H_10_N_4_O_3_ 271.0826, found 271.0825.

### 3.4. Synthesis of 7-Amino-5-phenylimidazo[1,5-b]pyridazin-2(1H)-one (***52***)

#### 3.4.1. Method A

Acid **51** (5 mmol) was dissolved in 5 mLof diphenyl ether under constant heating and stirring, and 15 mmol of copper (II) acetate were added. The reaction mixture was boiled for 3 h. The formed precipitate was filtered off, washed with water, and then recrystallized from a mixture of i-PrOH/DMF, 2:1.

#### 3.4.2. Method B

Ethyl propionate **53** (5 mmol) was added dropwise to 5 mmol of diaminoimidazole **44a**, dissolved in 5 mL of methanol in the presence of catalytic amounts of acetic acid (2–3 drops) under constant stirring and heating at 40 °C for 60 min. The formed precipitate was filtered off and recrystallized from a mixture of i-PrOH/DMF, 2:1.

#### 3.4.3. 7-Amino-5-phenylimidazo[1,5-*b*]pyridazin-2(1H)-one (**52**)

Yields (36.2 mg, 32% (method A) and 84.8 mg, 75% (method B)), yellow. Mp.: 271–273 °C. ^1^H NMR (500 MHz, DMSO-*d*_6_, δ ppm): 11. 35 (br. s, 1H), 8.08 (d, *J* = 9.7, 1H), 7.79 (d, *J* = 7.5, 2H), 7.38 (t, *J* = 7.6, 2H), 7.20 (t, *J* = 7.3, 1H), 6.08 (d, *J* = 9.6, 1H), 5.75 (s, 2H). ^13^C{^1^H} NMR (125.8 MHz, DMSO-*d*_6_, δ ppm): 159.0, 143.2, 134.9, 129.7, 128.6, 126.6, 125.8, 125.1, 115.7, 104.4. HRMS (ESI), *m*/*z*: [M+H^+^] calc. for C_12_H_10_N_4_O 227.0928, found 227.0926.

## 4. Conclusions

A new approach for the synthesis of 7-amino-1,2-dihydro-1-R-2-oxo-5-phenylimidazo[1,5-*b*] pyridazine-4-carboxylic acids and their esters (**45a**–**e**) from readily available 1,2-diamino-4-phenylimidazoles and dimethylacetylene dicarboxylate (DMAD) was suggested. DFT calculations indicatedthat this synthesis proceeds via the formation of low-barrier intermediates. The use of a protic solvent (methanol and few drops of acetic acid) accelerates the course of the processes; this factis probably not due to their involvement as catalysts, but to a change in the dielectric constant of the medium, able to stabilizepolar intermediates. DFT results confirm the experimental results, moreover, the observed course of the reaction (formation of compounds **45a**–**e**) is confirmed by spectroscopic results.

## Data Availability

Not applicable.

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
