# Peer review of "A Multifield Study on Dimethyl Acetylenedicarboxylate: A Reagent Able to Build a New Cycle on Diaminoimidazoles"

_molecules, 2022, doi:10.3390/molecules27103326_

Round 1

Reviewer 1 Report

Vandyshev, Spinelli and coworkers report an interesting physical organic study on anellation reactions of DMAD with binucleophilic heterocycles, a topic that has also found attention in exisiting literature. They take a computational approach to rationalize possible reaction pathways of formation. Some aspects, however, need attention prior to considering for publication.

1) In the introduction (Schemes 1-6) it would be helpful for the benign reader to give more detailed information on scope (if applicable) and certainly the yields of the obtained anellated products.

2) Some calculated energy barriers appear to be quite high with respect to the experimental data (e.g. reactions already proceed well in dichloromethane at 40 degrees).

3) In Scheme 10 the authors propose and calculate a stepwise, zwitterionic mechanism. To my opinion it also would make sense to consider a pericyclic aza-ene mechanism from 44a + 1 to 57 directly, which might have a considerably lower barrier of activiation.

Especially, the quite significant deviation of the calculated data in the intrinsic solvent model need special attention and presumably the consideration of alternative pathways, which might proceed with lower calculated barriers.

Author Response

Vandyshev, Spinelli and coworkers report an interesting physical organic study on anellation reactions of DMAD with binucleophilic heterocycles, a topic that has also found attention in exisiting literature. They take a computational approach to rationalize possible reaction pathways of formation. Some aspects, however, need attention prior to considering for publication.

  1. In the introduction (Schemes 1-6) it would be helpful for the benign reader to give more detailed information on scope (if applicable) and certainly the yields of the obtained anellated products.

In the introduction, schemes 1-6 are amended accordingly: substituents are listed and yields of the target annelated products are added

  1. Some calculated energy barriers appear to be quite high with respect to the experimental data (e.g. reactions already proceed well in dichloromethane at 40 degrees).

Yes, technically the barriers of some stages look quite high. However, we would like to remind you that barriers up to 20 kcal/mol are easily overcome even at low temperatures. This allows, for example, tautomerism processes to be studied by NMR methods. (https://doi.org/10.1002/ejoc.201402692). Barriers on the order of 35 kcal/mol can allow intermolecular reactions to take place at room temperature, which does not contradict the experiment.

  1. In Scheme 10 the authors propose and calculate a stepwise, zwitterionic mechanism. To my opinion it also would make sense to consider a pericyclic aza-ene mechanism from 44a + 1 to 57 directly, which might have a considerably lower barrier of activiation.

The pericyclic aza-ene mechanism from 44a + 1 to 57 directly was investigated.  However, the localisation of the transition states is not possible. Synchronous hydrogen shift and C-C bond formation do not occur.

Especially, the quite significant deviation of the calculated data in the intrinsic solvent model need special attention and presumably the consideration of alternative pathways, which might proceed with lower calculated barriers.

Indeed, the solvent involvement at the molecular level shows the potential involvement of the solvent in the 44a + 1 reaction. It is possible that in our case the process involves solvent clusters. However, the study of such processes is not of interest in terms of the mechanism of the 44a + 1 process under study. The processes in clusters are accompanied by a multistage cascade of subtle transformations in the cluster structure and, in fact, are reduced to the chemistry of the cluster, but not to the chemistry of the process studied in our paper.

Reviewer 2 Report

This manuscript described the reaction of five different 1,2-diamino-4-phenylimidazole derivatives with dimethyl acetylenedicarboxylate (DMAD). Different possible products were proposed for this reaction and among these the structure of the products obtained was identified based on the spectroscopic data. Quantum chemical calculations were performed and the minimum energy paths for the formation of the proposed possible products were reported, both for the bimolecular pathways of interaction of 4-phenyl-1H-imidazole-1,2-diamine and DMAD and for the trimolecular processes (including methanol or acetic acid). These data are consistent with the experimental results.

The introduction section requires an extensive review, as there are some inaccuracies and errors. Please, check this section carefully and consider the following indications.

-line 53: the structure 4a does not match the name “dihydro[1,2,4]triazolo[4,3-a]pyrimidinones” and is different from the one shown in ref 21

-line 54: in ref 21 antibacterial activity of the molecules is not reported

-Scheme 1: compound 7 should be a nitroimidazole; compound 11 was obtained in toluene, not MeOH; compound 13 and the name reported in line 61 (“[1,3]thiazino[3,2-a]indolone”) does not match with the structures of ref 27

-in ref 25 I can not find the corresponding structures and conditions

-Scheme 2: in the cited references, compounds 18 were obtained in MeOH, EtOH or toluene, not CH3CN

-line 84 and Scheme 4: the reactions reported in ref 34 are with diethyl acetylenedicarboxylate and not DMAD

-line 92: the ref for the reaction with compounds 30 is missed

-line 94 and Scheme 5: the reactions reported in ref 35 with both compounds 31 or 32 are in MeOH and rt, while the subsequent cyclizations are in diphenyl ether at rf

In the materials and methods (line 390) specify the gradient elution used for HPLC/MS analyses.

Minor notes:

-line 48: add a comma between “substrates” and “depend

-line 51-52: change in “such as amino- and mercaptotriazoles 2a and 2b, respectively

-line 53-54: the numbers referred to the structures should be 4a and 4b (not 4 and 5)

-line 65: remove the comma between “DMAD” and “can

-line 102: “benzalideneamino” correct in “benzylidenamino”

-line 104-105: correct the name of compound 41

-line 106: “2-amino-4-aryl-1-arylideneaminolimidazoles” correct in “2-amino-4-aryl-1-arylideneaminoimidazoles

-line 108-109: correct the name of compound 43

-line 132: “the spectra of the reaction masses” may be “the mass spectra of the reactions

-Table 1: in the caption and in the table columns specify that the content of 1,2-diamino-4-phenylimidazole is a %. In the caption specify the reaction conditions (amount of reagent, reflux). In the caption “the reaction mass” may be “HPLC-MS analyses of the reactions”. In the table standardize the value approximation.

-line 143: could be change to “In this case, we have observed…

-line 146: change “reactants” in “reactant

-line 147: remove “the reaction in

-line 181: add a comma between “imidazole ring” and “as

-line 182 and 202: correct “1,2-diaminoimidazles” in “1,2-diaminoimidazoles

-line 220, 284, 292: correct “4-phenyl-1H-imidazole-1,2-diamine” with the “H” in italics

-line 233: correct “TS2” in “TS3

-line 239: change “After we have considered the reaction” in “Then we considered the reaction

-line 240, 256: add the reference also to Scheme 7

-line 265: add “(PES)” after “potential energy surface”. Please, add consideration about the energy of this process

-line 276: “As in the three previous cases” correct in “As in the two previous cases

-line 287: remove the comma between “bimolecular processes” and “showed

-line 289: correct “adduct 33” in “adduct 54

-line 303: remove the comma between “led to 63” and “was

-line 308: change in “the covalent adduct 63

-line 348: “as in the case of the participation of methanol, was higher in energy during all stages than in the corresponding bimolecular processes” in the case of methanol, it is not exactly true: if the sentence refers to the TS, TS11 is lower in energy than TS2, if it is referred to the intermediate it is true only for 62 (vs 54).

-line 352: “than the infinitely separated intermediates and the solvent molecule” could be “than the intermediates infinitely separated from the solvent molecules

-line 356: change in “In fact, the higher the polarity of the medium, the higher the stabilization…

-line 360: “see accompanying materials” change in “see supporting materials

-line 385-386: superscript the numbers in "1H", "13C" and subscript the number in "d6"

-line 435: correct in “C21H17ClN4O3

-line 438: add “-“ between “dihydro” and “2

-line 453: the signal at 20.5 is miss in the 13C NMR

-line 484: correct in “C12H10N4O

-line 494: “able to stabilizing” change in “able to stabilise

In the supporting information:

-correct the first title both in the table of content and in p. S1 (it referred to Scheme 11 and 12)

-p. S2: “covalent adduct 61 from structure 47” correct in “covalent adduct 61 from structure 66

-improve the image quality of the NMR spectra and add the name, the structure and the number of the corresponding molecules

Author Response

This manuscript described the reaction of five different 1,2-diamino-4-phenylimidazole derivatives with dimethyl acetylenedicarboxylate (DMAD). Different possible products were proposed for this reaction and among these the structure of the products obtained was identified based on the spectroscopic data. Quantum chemical calculations were performed and the minimum energy paths for the formation of the proposed possible products were reported, both for the bimolecular pathways of interaction of 4-phenyl-1H-imidazole-1,2-diamine and DMAD and for the trimolecular processes (including methanol or acetic acid). These data are consistent with the experimental results.

The introduction section requires an extensive review, as there are some inaccuracies and errors. Please, check this section carefully and consider the following indications.

  1. line 53: the structure 4a does not match the name “dihydro[1,2,4]triazolo[4,3-a]pyrimidinones” and is different from the one shown in ref 21

Corrected as per observation

  1. line 54: in ref 21 antibacterial activity of the molecules is not reported

Corrected as per observation and lost links added

  1. Scheme 1: compound 7 should be a nitroimidazole; compound 11 was obtained in toluene, not MeOH; compound 13 and the name reported in line 61 (“[1,3]thiazino[3,2-a]indolone”) does not match with the structures of ref 27

Corrected as per observation

  1. in ref 25 I can not find the corresponding structures and conditions

In the text of the relevant patent (reference number after revision - 38), on page 80, the conditions given in the relevant scheme 5 are reflected

  1. Scheme 2: in the cited references, compounds 18 were obtained in MeOH, EtOH or toluene, not CH3CN

Corrected as per observation

  1. line 84 and Scheme 4: the reactions reported in ref 34 are with diethyl acetylenedicarboxylate and not DMAD

Corrected as per observation

  1. line 92: the ref for the reaction with compounds 30 is missed

Corrected as per observation

  1. line 94 and Scheme 5: the reactions reported in ref 35 with both compounds 31 or 32 are in MeOH and rt, while the subsequent cyclizations are in diphenyl ether at rf

Corrected as per observation

  1. In the materials and methods (line 390) specify the gradient elution used for HPLC/MS analyses.

The necessary information has been entered in the "materials and methods" section

Minor notes:

  1. line 48: add a comma between “substrates” and “depend

Corrected as per observation

  1. line 51-52: change in “such as amino- and mercaptotriazoles 2a and 2b, respectively

Corrected as per observation

  1. line 53-54: the numbers referred to the structures should be 4a and 4b (not 4 and 5)

Corrected as per observation

  1. line 65: remove the comma between “DMAD” and “can

Corrected as per observation

  1. line 102: “benzalideneamino” correct in “benzylidenamino”

Corrected as per observation

  1. line 104-105: correct the name of compound 41

Corrected as per observation

  1. line 106: “2-amino-4-aryl-1-arylideneaminolimidazoles” correct in “2-amino-4-aryl-1-arylideneaminoimidazoles

Corrected as per observation

  1. line 108-109: correct the name of compound 43

The title given in this paper has been borrowed from the original article (Reference 42)

  1. line 132: “the spectra of the reaction masses” may be “the mass spectra of the reactions

Corrected as per observation

  1. Table 1: in the caption and in the table columns specify that the content of 1,2-diamino-4-phenylimidazole is a %. In the caption specify the reaction conditions (amount of reagent, reflux). In the caption “the reaction mass” may be “HPLC-MS analyses of the reactions”. In the table standardize the value approximation.

Corrected as per observation

  1. line 143: could be change to “In this case, we have observed…

Corrected as per observation

  1. line 146: change “reactants” in “reactant

Corrected as per observation

  1. line 147: remove “the reaction in

Corrected as per observation

  1. line 181: add a comma between “imidazole ring” and “as

Corrected as per observation

  1. line 182 and 202: correct “1,2-diaminoimidazles” in “1,2-diaminoimidazoles

Corrected as per observation

  1. line 220, 284, 292: correct “4-phenyl-1H-imidazole-1,2-diamine” with the “H” in italics

Corrected as per observation

  1. line 233: correct “TS2” in “TS3

Corrected as per observation

  1. line 239: change “After we have considered the reaction” in “Then we considered the reaction

Corrected as per observation

  1. line 240, 256: add the reference also to Scheme 7

Corrected as per observation

  1. line 265: add “(PES)” after “potential energy surface”. Please, add consideration about the energy of this process

Corrected as per observation

  1. line 276: “As in the three previous cases” correct in “As in the two previous cases

Corrected as per observation

  1. line 287: remove the comma between “bimolecular processes” and “showed

Corrected as per observation

  1. line 289: correct “adduct 33” in “adduct 54

Corrected as per observation

  1. line 303: remove the comma between “led to 63” and “was

Corrected as per observation

  1. line 308: change in “the covalent adduct 63

Corrected as per observation

  1. line 348: “as in the case of the participation of methanol, was higher in energy during all stages than in the corresponding bimolecular processes” in the case of methanol, it is not exactly true: if the sentence refers to the TS, TS11 is lower in energy than TS2, if it is referred to the intermediate it is true only for 62 (vs 54).

Corrected as per observation

  1. line 352: “than the infinitely separated intermediates and the solvent molecule” could be “than the intermediates infinitely separated from the solvent molecules

Corrected as per observation

  1. line 356: change in “In fact, the higher the polarity of the medium, the higher the stabilization…

Corrected as per observation

  1. line 360: “see accompanying materials” change in “see supporting materials

Corrected as per observation

  1. line 385-386: superscript the numbers in "1H", "13C" and subscript the number in "d6"

Corrected as per observation

  1. line 435: correct in “C21H17ClN4O3

Corrected as per observation

  1. line 438: add “-“ between “dihydro” and “2

Corrected as per observation

  1. line 453: the signal at 20.5 is miss in the 13C NMR

Corrected as per observation

  1. line 484: correct in “C12H10N4O

Corrected as per observation

  1. line 494: “able to stabilizing” change in “able to stabilise

Corrected as per observation

In the supporting information:

  1. correct the first title both in the table of content and in p. S1 (it referred to Scheme 11 and 12)

Corrected as per observation

  1. S2: “covalent adduct 61 from structure 47” correct in “covalent adduct 61 from structure 66

Corrected as per observation

  1. improve the image quality of the NMR spectra and add the name, the structure and the number of the corresponding molecules

NMR spectra images have been replaced with better ones, according to your recommendations

Round 2

Reviewer 1 Report

The authors have largely addressed the raised comments and queries. The energy barriers are rather in the range of 30-40 kcal/mol, which would rather contradict with room temp experiments. We are talking about reaction times at room temp of 2 h and not 12 h or more. Certainly, the experiment is the outset and it proceeds as described. However, computational chemists should realistically keep in mind that the chosen model (calculations are nothing else than models) should reflect the experimental results cum grano salis. Tautomerisms are either unimolecular or occur to low tunneling barriers via proton transfer. Here, yet the initial steps are clearly second order. Yet, since this is not a predominant computational work but rather an experimental study, it is suggested to move the calculations to the supp inf and indicate in a few explaining sentences that the tendency is roughly suggesting a suggested mechanistic rationale. Harder evidence would certainly come from kinetic measurements, which should be doable with these simple systems. The scientific value would also increase beyond plain synthesis.